# Mitochondrial targeted meganuclease as a platform to eliminate mutant mtDNA in vivo

Ugne Zekonyte[1], Sandra R. Bacman[2], Jeff Smith[3], Wendy Shoop[3], Claudia V. Pereira[2], Ginger Tomberlin[3], James Stewart [4,5], Derek Jantz[3] & Carlos T. Moraes[2]✉

Diseases caused by heteroplasmic mitochondrial DNA mutations have no effective treatment or cure. In recent years, DNA editing enzymes were tested as tools to eliminate mutant mtDNA in heteroplasmic cells and tissues. Mitochondrial-targeted restriction endonucleases, ZFNs, and TALENs have been successful in shifting mtDNA heteroplasmy, but they all have drawbacks as gene therapy reagents, including: large size, heterodimeric nature, inability to distinguish single base changes, or low flexibility and effectiveness. Here we report the adaptation of a gene editing platform based on the *I-CreI* meganuclease known as ARCUS®. These mitochondrial-targeted meganucleases (mitoARCUS) have a relatively small size, are monomeric, and can recognize sequences differing by as little as one base pair. We show the development of a mitoARCUS specific for the mouse m.5024C>T mutation in the mt-tRNA^Ala gene and its delivery to mice intravenously using AAV9 as a vector. Liver and skeletal muscle show robust elimination of mutant mtDNA with concomitant restoration of mt-tRNA^Ala levels. We conclude that mitoARCUS is a potential powerful tool for the elimination of mutant mtDNA.

[1] Graduate Program Human Genetics and Genomics, University of Miami Miller School of Medicine, Miami, FL, USA. [2] Department of Neurology, University of Miami Miller School of Medicine, Miami, FL, USA. [3] Precision BioSciences, Durham, NC, USA. [4] Faculty of Medical Sciences, Biosciences Institute, Newcastle University, Newcastle upon Tyne, UK. [5] Wellcome Centre for Mitochondrial Research, Faculty of Medical Sciences, Newcastle University, Newcastle upon Tyne, UK. ✉email: Cmoraes@med.miami.edu

Mitochondrial DNA (mtDNA) is a small 16.5 kb circular molecule that encodes for 37 genes: 13 subunits of the oxidative phosphorylation system, as well as the 22 transfer RNA (tRNA)'s and 2 ribosomal RNA's needed for mitochondrial translation[1]. On average, human cells contain ~1000 mtDNA molecules[2]. Mutant and wild-type (WT) mtDNA molecules can co-exist within the cell, a phenomenon called mtDNA heteroplasmy. Specific threshold levels of mutant mtDNA need to be reached, to compromise cell function and for disease to manifest[2]. This is also an important aspect in mtDNA gene therapies, as not all mutant mtDNA need to be eliminated to see improvements in symptoms.

The prevalence of mtDNA diseases in the population has been estimated at 1 : 5000 individuals[3]. To date, over 250 pathogenic mutations in mtDNA have been identified[4]. MtDNA diseases are multi-systemic disorders that present very heterogenous symptoms, even in patients carrying the same mutation, including but not limited to optic atrophy, stroke-like episodes, cardiomyopathies, muscle weakness, and neurodegeneration. However, essentially any organ in the body can be affected[5].

Current mtDNA gene therapy developments take advantage of the mitochondria's lack of double-stranded break (DSB) repair mechanism. After mtDNA is cleaved and linearized, it is quickly degraded in the cell, followed by repopulation of mtDNA to original levels by replicating residual mtDNA[6]. We and others have used restriction endonucleases[7–12], zinc finger nucleases (mitoZFNs)[13–16], transcription activator-like effector nucleases (mitoTALENs)[17–20], and mitoTev-TALE[21] to cleave and eliminate mutant mtDNA.

The ARCUS gene-editing platform, developed by scientists at Precision BioSciences, is based on the homing endonuclease I-CreI, which comes from the *Chlamydomonas reinhardtii* chloroplast genome, and is part of the LAGLIDADG motif meganuclease family[22]. I-CreI is a homodimeric enzyme that associates with a palindromic 22 bp double-stranded DNA sequence, where it produces a DSB[23]. Previously, these naturally homodimeric enzymes were engineered into monomers by using a peptide linker. In silico and directed evolution allowed the targeting of almost any DNA sequence[24–27]. These meganucleases' small size (1092 bp/40 kDa), monomeric nature, and specificity[24–27] make them particularly promising for mtDNA editing. Here we designed a mitoARCUS specific to the m.5024C>T tRNA^Ala mutation, which was delivered with an adeno-associated virus (AAV9), to target to a wide range of tissues, including the skeletal muscle, heart, and liver[28], resulting in a specific elimination of mtDNA and normalization of mt-tRNA^Ala levels.

## Results

**Designing a mitoARCUS against the m.5024C>T sequence in the mtDNA-encoded mt-tRNA^Ala gene.** Using Precision BioSciences' protein engineering pipeline, a candidate meganuclease recognizing the mutated (m.5024C>T) mouse mtDNA sequence [5′-ATAAGGA**T**TGTAAGACTTCATC-3′] was produced. Analysis of the designer nuclease using a green fluorescent protein (GFP)-based DSB recombination assay in Chinese Hamster Ovary (CHO) cells showed that the specific ARCUS nuclease (termed MIT 11–12) yielded ~80% GFP+ cells when tested against the intended (mutant) target sites but <5% GFP+ cells when tested against the WT target sites (Fig. 1a). For ex vivo expression, two variants were designed, CF and CSF, which differed by the mitochondrial localization sequence (MLS). We added either a Cox8[29] (construct CF) or a Cox8/Su9[20,21] (construct CSF) MLS to the N terminus. We also added a FLAG tag between the MLS and the meganuclease (Fig. 1b). Transient transfection in HeLa cells showed that both mitoARCUS constructs made proteins that

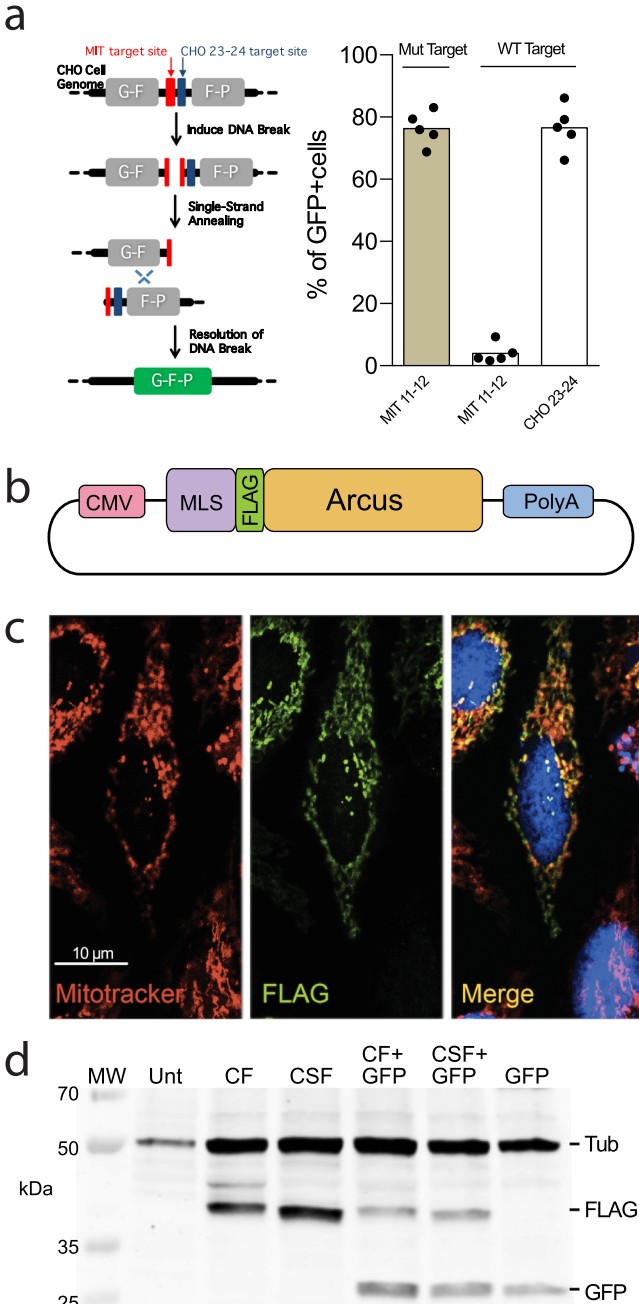

localized to mitochondria (Fig. 1c shows a representative cell for the CSF construct). A western blot confirmed that both CF and CSF constructs expressed the expected size proteins, after transfection of HEK293T cells (Fig. 1d). As the pattern in the western blot showed no spurious bands in the CSF-transfected cells, we chose the CSF variant for the subsequent experiments.

**MitoARCUS effectively shifted mtDNA heteroplasmy in cultured cells.** The efficiency of the mitoARCUS was first tested in mouse embryonic fibroblasts (MEFs) derived from a heteroplasmic mouse model carrying the mtDNA m.5024C>T mutation[20]. Experiments were conducted in cells carrying 50% of m.5024C>T mutation. As transfection efficiency was relatively low, we co-transfected a mitoARCUS-expressing plasmid with a plasmid expressing GFP (2 mitoARCUS : 1 GFP ratio). GFP-positive cells comprised 11–20% of the total cell population after

**Fig. 1 mitoARCUS construct and mitochondrial expression. a** For an MIT nuclease (against the mouse mtDNA tRNA-Ala mutation [MIT 11–12]), a pair of engineered CHO lines were produced to carry either the wild-type (WT) or mutant mtDNA target site in the nuclear DNA. The target site was positioned between direct repeats of a GFP gene such that cleavage of the target site promotes homologous recombination events between repeated regions to yield a functional GFP. In addition, there is a target site for a positive control nuclease ("CHO 23-24") incorporated next to the MIT target site (left panel). Each of the cell lines was transfected with mRNA encoding MIT 11–12 or CHO 23-24 (control) and cells were assayed by flow cytometry 2, 5, 7, 9, and 12 days post transfection for the percentage of GFP + cells (the average GFP fluorescence for the different time points is shown in the right panel). As the different time points are not biological replicates, no statistic was applied. **b** mitoARCUS gene construct for ex vivo expression includes CMV promoter, mitochondrial localization sequence (MLS) of Cox8 or Cox8/Su9, Flag tag for immunological detection, Meganuclease (ARCUS) sequence, and PolyA tail. **c** Immunofluorescence done on HeLa cells 24 h after transfection with mitoARCUS. MitoTracker stains mitochondria red, Flag stains mitoARCUS green, and merged image shows colocalization (yellow) of mitoARCUS to mitochondria. Images taken at ×40 magnification. This experiment was repeated twice with identical results. (**d**) Western blot depicting mitoARCUS expression (FLAG) in HEK293T cells 24 h after transfection with either CF or CSF construct. Lanes CF + GFP and CSF + GFP depict protein expression in cells transfected with mitoARCUS constructs in which we added a GFP sequence. Lane Unt represents untransfected cells. Lane GFP represents cells transfected with GFP only. Tubulin (Tub) expression was used as a loading control. This experiment was performed once.

transfection. Co-transfected cells were fluorescence-activated cell (FACS) sorted 24 h later into "Black" populations (not transfected) and "Green" populations (green fluorescence, transfected) (Fig. 2a). We used PCR/restriction fragment length polymorphism (RFLP) to determine mtDNA heteroplasmy changes in the "Black" and "Green" cell populations of two independent experiments (Fig. 2b). Quantification showed that there was a large shift (50–60%) in heteroplasmy in the "Green" population when compared to the "Black" population. Cells transfected only with the GFP plasmid did not show changes in heteroplasmy. Not surprisingly, there was a small shift in the "Black" cell populations (10–20%) that can be explained by some cells in the "Black" population not having incorporated GFP co-transfectant plasmid, but likely did incorporate the mitoARCUS plasmid (Fig. 2c).

To determine the biological significance of heteroplasmic change, we used an MEF cell line that harbored high levels (90%) of mutant mtDNA[20]. We co-transfected cells with plasmids expressing mitoARCUS and GFP, and sorted them as described above. mtDNA was analyzed 1, 7, 14, and 21 days after transfection by PCR/RFLP. Again, GFP-positive cells comprised 11–20% of the total cell population after transfection. There was a significant shift in heteroplasmy (~25%) in "Green" cells at 24 h post transfection that was maintained over a 2-week period (Fig. 2d, e). As expected, there was a depletion of total mtDNA levels in the "Green" cell populations 24 h after transfection, which then returned to normal levels after 21 days (Fig. 2f). The "Black" population had a very mild decrease, likely because some cells received the mitoARCUS, as previously explained (Fig. 2f). Cells grown for 3 weeks were analyzed for their oxygen consumption rate (OCR). We found that untransfected cells had impaired respiration compared to WT controls. On the other hand, "Green" cell populations had significantly improved OCR (Fig. 2g). We also observed a small improvement in OCR in the "Black" cells (Fig. 2g), which is not surprising because of the mild change in heteroplasmy discussed above.

**In vivo administration of mitoARCUS.** MitoARCUS was cloned into an AAV9 vector to produce viral particles for in vivo applications. AAV9-mitoARCUS was administered systemically via retro-orbital injection in heteroplasmic mice carrying the m.5024C>T mutation[30]. AAV9-GFP was used as a control in a different group of heteroplasmic mice. Toe biopsies were used to determine starting levels of heteroplasmy prior to injection and showed varying levels of heteroplasmy between animals, but most were around 50–80% mutant. We injected juvenile animals at 2.5 weeks of age, as well as adult animals at 6 weeks of age. The injected animals were killed 6, 12, and 24 weeks post injection (PI), and the heart, tibialis anterior (TA), quadriceps, gastrocnemius, kidney, liver, brain, and spleen were collected for analysis. In addition, we injected another group of mice at 2.5 weeks and killed them at 5 and 10 days PI for the analysis of early events.

**MitoARCUS effectively shifted mtDNA heteroplasmy in juvenile animals.** Heteroplasmic m.5024C>T mice were injected systemically with AAV9-mitoARCUS at 2.5 weeks of age. AAV9-mitoARCUS-injected animals showed consistent expression of mitoARCUS in the heart, skeletal muscles, and sometimes the liver at all time points: 6, 12, and 24 weeks PI, as shown by western blot analysis using a Flag antibody against the construct (Fig. 3a). AAV9-GFP-injected animals showed GFP expression in the same tissues, although with higher expression in the liver at 6 and 12 weeks PI (Fig. 3a). Corresponding RFLP analysis showed a significant decrease in mutant mtDNA in the liver and TA at 6 weeks PI in AAV9-mitoARCUS-injected animals (Fig. 3a). Over time, heteroplasmy shifts became greater, and at 24 weeks PI, there was a significant decrease in mutant mtDNA in the heart, skeletal muscles, kidney, and liver (Fig. 3b). AAV9-GFP-injected controls had similar levels of mutation across all tissues after injection (Fig. 3a, b). Brain was used as a negative control to normalize changes in heteroplasmy, because no expression was observed in the brain after injection of either AAV9-mitoARCUS or AAV9-GFP (Fig. 3a). There was no significant depletion of mtDNA levels seen in any of the analyzed tissues at 6, 12, or 24 weeks PI (Fig. 3c), demonstrating that there was no major nonspecific mtDNA effects. Mice weights did not differ between mitoARCUS-treated and control animals at all time points (Supplementary Fig. 1a–f).

As, despite the major heteroplasmy shift, there was no clear expression of the mitoARCUS in the liver at 6, 12, or 24 weeks PI, we injected additional animals at 2.5 weeks and killed them 5 and 10 days PI. At 5 and 10 days PI, mitoARCUS expression was only detected in the liver (Supplementary Figs. 2a and 3a). However, GFP expression was also visible in the heart, skeletal muscles, and liver, possibly because of higher expression of this construct or a better antibody (Supplementary Figs. 2a and 3a). Accordingly, heteroplasmy changes at 5 and 10 days PI were only observed in liver (Supplementary Figs. 2b and 3b). There was an average of 22% reduction in mutant mtDNA at 5 days PI (Supplementary Fig. 2c) and 55% at 10 days PI (Supplementary Fig. 3c). To determine whether apoptosis played a role in these changes, we analyzed the expression of Proliferating Cell Nuclear Antigen (PCNA a marker of liver regeneration), Caspase-3, and cleaved Caspase3 in liver samples. We observed no differences between mitoARCUS-treated animals and controls at both time points (Supplementary Figs. 2d and 3d). We observed similar levels of uncleaved caspase-3 in treated and control animals, but not cleaved caspase-3 (Supplementary Figs. 2d and 3d). In addition, we did not observe depletion of total mtDNA levels at 5 or 10 days PI in the liver or TA (Supplementary Figs. 2e and 3e). Furthermore, hematoxylin and eosin staining of the liver did not show any morphological differences between

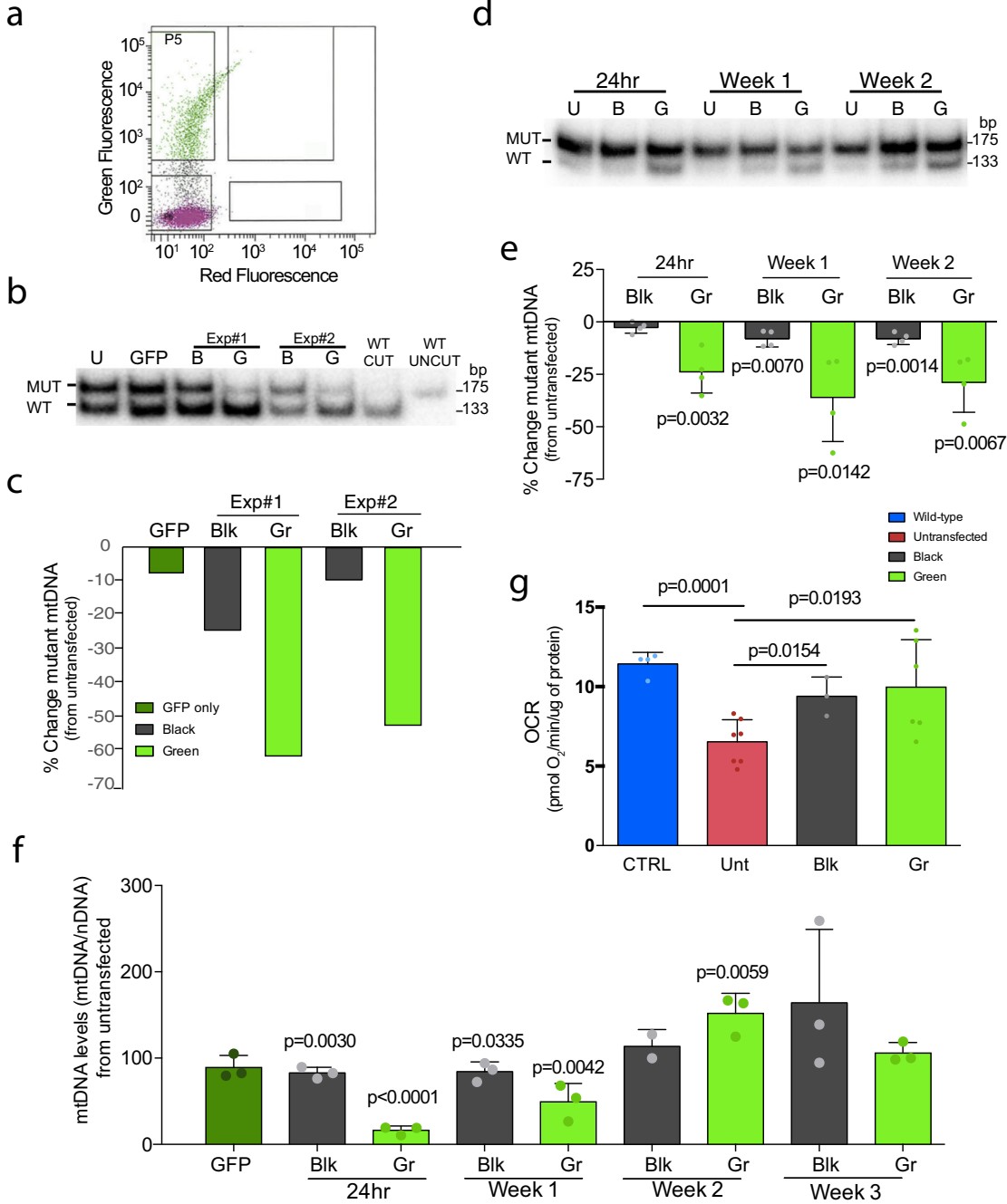

**Fig. 2 mitoARCUS effect on heteroplasmic cells carrying the tRNAAla mutation (m.5024C>T). a** Example of FACS cell sorting gating. Cells were sorted by the presence of GFP co-expression: "Black" cells (bottom gate) and "Green" cells (top gate). **b** RFLP-HOT PCR analysis of two independent transfections and cell-sorting experiments of heteroplasmic cells carrying 50% m.5024C>T mutation. Mutant levels in the Green cell populations (Gr) were compared to Untransfected cells (U). This experiment was done once. **c** Quantification of heteroplasmy shift from the two cell-sorting experiments in cells carrying 50% mutation described in **b**. Results were compared to Untransfected cell heteroplasmy. **d** RFLP "last-cycle hot" PCR analysis of heteroplasmic cells carrying high heteroplasmic mutant load (90%) transfected with mitoARCUS over time. This experiment was repeated three times with similar results. **e** Quantification of Fig. 2d. Values are normalized to Untransfected cells. Black cells are named Blk ($n = 4$). **f** Total mtDNA levels were checked in highly mutant cells transfected with mitoARCUS and compared to untransfected cells 24 h after transfection, and followed for 3 weeks after transfection ($n = 3$). p values are related to untransfected cells (100%). **g** Oxygen consumption rate (OCR) was deduced in cells carrying high levels of heteroplasmic mutant mtDNA that were transfected with mitoARCUS and grown for 3 weeks [$n = 3$ (CTRL), $n = 7$ (Unt), $n = 3$ (Blk), $n = 7$ (Gr)]. Data are mean ± SEM. Statistical analysis was performed using two-tailed Student's $t$-test.

AAV9-mitoARCUS-injected, AAV9-GFP-injected, and non-injected controls (Supplementary Fig. 4).

Even though mitoARCUS localized exclusively to the mitochondria, we tested whether nuclear off-target editing occurred. To do so, targeted amplicon sequencing was performed on a selection of the sites that were identified from an in vitro, genome-wide, unbiased off-targeting assay based on GUIDE-Seq[31] (see "Methods"). DNA from the TA and liver tissues from the young mice killed at 24 weeks PI were evaluated by targeted amplicon sequencing. This method of analysis detects any genetic

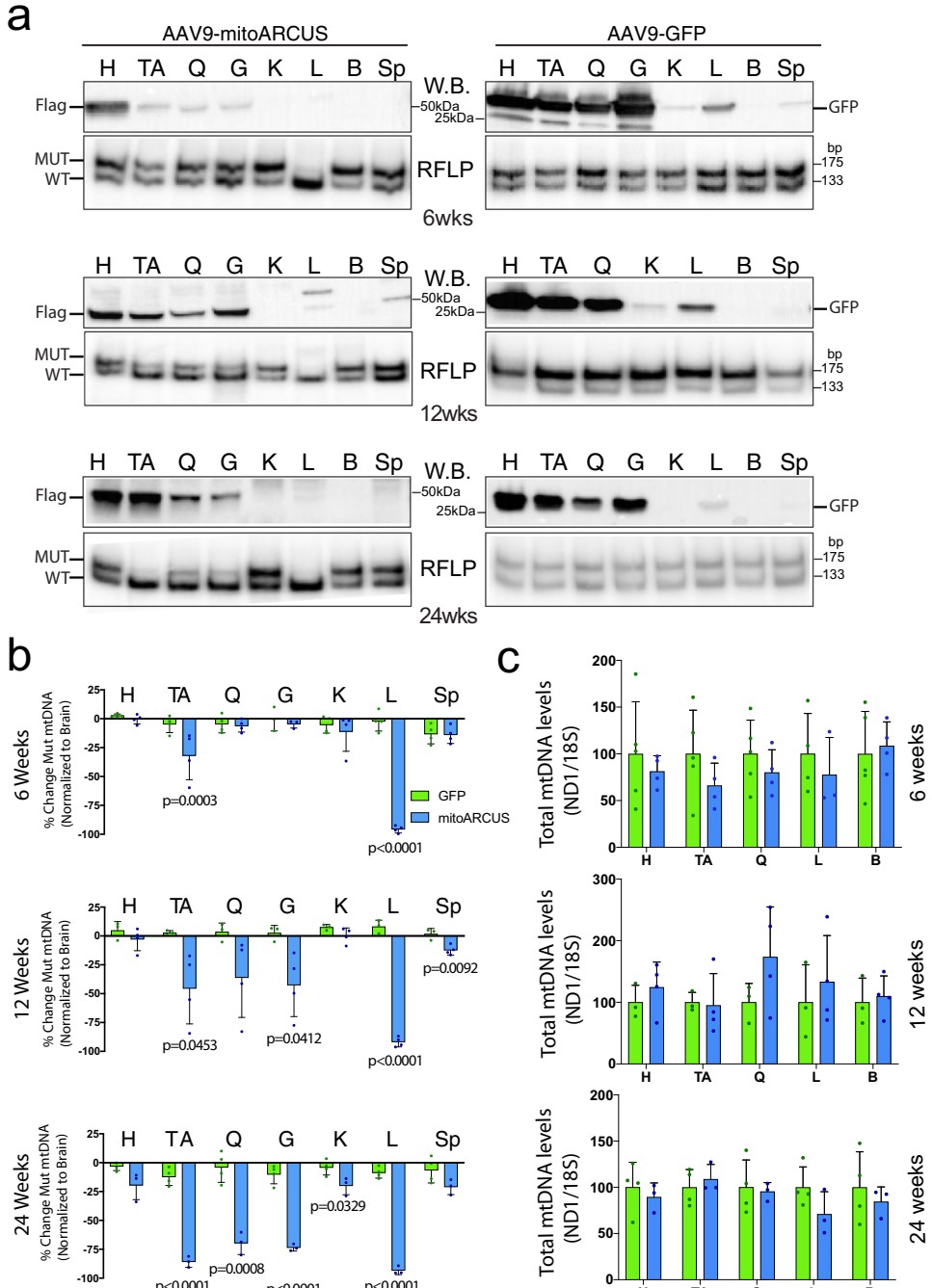

**Fig. 3 AAV9-mitoARCUS effect in treated juvenile mice. a** Representative western blottings (W.B.) of homogenates (top panels) with Flag antibody for AAV9-mitoARCUS samples and GFP antibody for AAV9-GFP samples. RFLP "last-cycle hot" PCR analysis (RFLP, bottom panels) of DNA samples from the same injected animals at 6, 12, and 24 weeks PI. Similar analyses were performed for each animal. **b** Quantification of heteroplasmy shift shown as a percent change in heteroplasmy across all tissues at 6, 12, and 24 weeks PI normalized to brain tissue. Heteroplasmy levels of the heart (H), tibialis anterior (TA), quadriceps (Q), gastrocnemius (G), kidney (K), liver (L), and spleen (Sp) were compared to that of the brain (B) (negative for expression of mitoARCUS). **c** Quantification by RT-PCR of total mtDNA levels in the skeletal muscle, liver, and brain at 6 and 24 weeks PI using ND1 and ND5 mitochondrial primer/probes normalized to 18S (nuclear DNA). Data are mean ± SEM of $n = 4$ (with exception of 6 weeks GFP ($n = 5$) and 24 weeks mitoARCUS ($n = 3$)). Statistical analysis was performed using two-tailed Student's $t$-test.

variation within the amplicon, such as an indel. No indels were detected at any of the sites analyzed for any of the animals.

**MitoARCUS effectively shifted mtDNA heteroplasmy in adult animals.** Heteroplasmic m.5024C>T mice were also injected systemically with AAV9-mitoARCUS at 6 weeks of age, an age

that is less permissive to AAV-mediated expression compared to 2.5 weeks. Still, a strong expression was observed in the heart and liver, with weaker expression in skeletal muscles (Fig. 4a). AAV9-GFP-injected mice showed strong expression in the heart, TA, and liver, with weaker expression in quadriceps and gastrocnemius (Fig. 4a). RFLP analysis showed an essentially complete elimination of mutant mtDNA in the liver as early as 6 weeks

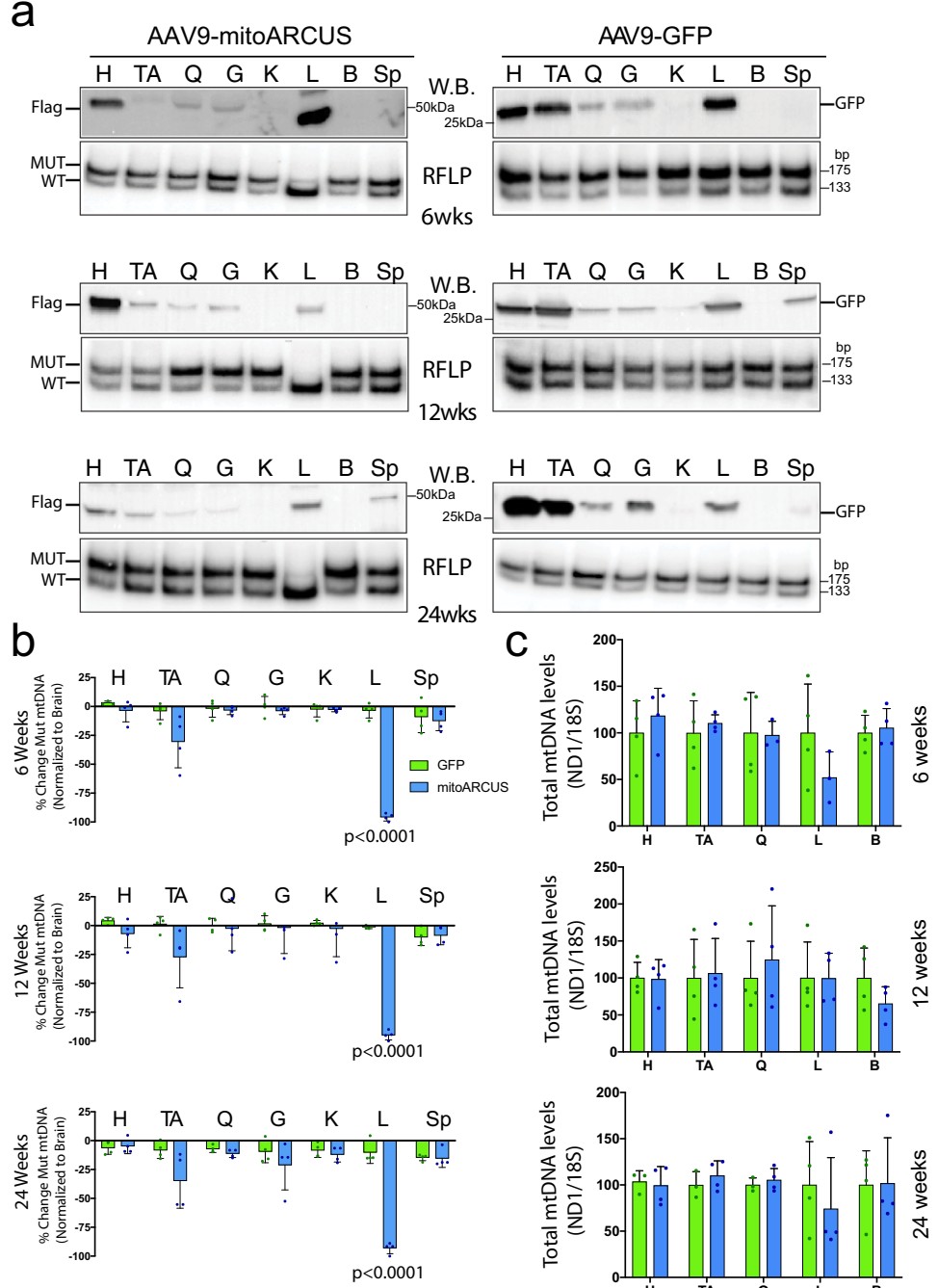

**Fig. 4 AAV9-mitoARCUS effect in treated adult mice. a** Representative western blottings (W.B.) of homogenates (top panels) with Flag antibody for AAV9-mitoARCUS samples and GFP antibody for AAV-GFP samples. RFLP "last-cycle hot" PCR analysis (RFLP, bottom panels) of DNA samples from the same injected animals, at 6, 12, and 24 weeks PI. Similar analyses were performed for each animal. **b** Quantification of heteroplasmy shift shown as percent change in heteroplasmy across all tissues at 6, 12, and 24 weeks PI normalized to brain tissue. Heteroplasmy of the heart (H), tibialis anterior (TA), quadriceps (Q), gastrocnemius (G), kidney (K), liver (L), and spleen (Sp) were compared to that of the brain (negative for expression of mitoARCUS). **c** Quantification by qPCR of total mtDNA levels were measured in the heart, skeletal muscle, liver, and brain (B) at 6 and 24 weeks PI using ND1 and ND5 mitochondrial primer/probes normalized to 18S (nuclear DNA). Data are mean ± SEM of $n = 4$. Statistical analysis was performed using two-tailed Student's $t$-test.

after injection in mitoARCUS-treated animals, which persisted over time (Fig. 4b). Some skeletal muscles showed a trend in decreasing mutant mtDNA (TA, gastrocnemius at 24 weeks PI), but results did not reach significance (Fig. 4b). AAV9-GFP-injected animals showed no change in the levels of heteroplasmy across all tissues (Fig. 4a, b). Brain was used to normalize the changes. Total mtDNA levels were slightly decreased in the liver

6 weeks PI of the AAV9-mitoARCUS-treated animals, but not after 12 or 24 weeks PI (Fig. 4c).

**Mt-tRNA^Ala levels were restored in the liver of mice treated with AAV9-mitoARCUS.** Mt-tRNA^Ala levels are decreased in tissues with high levels of mutant mtDNA in the m.5024C>T

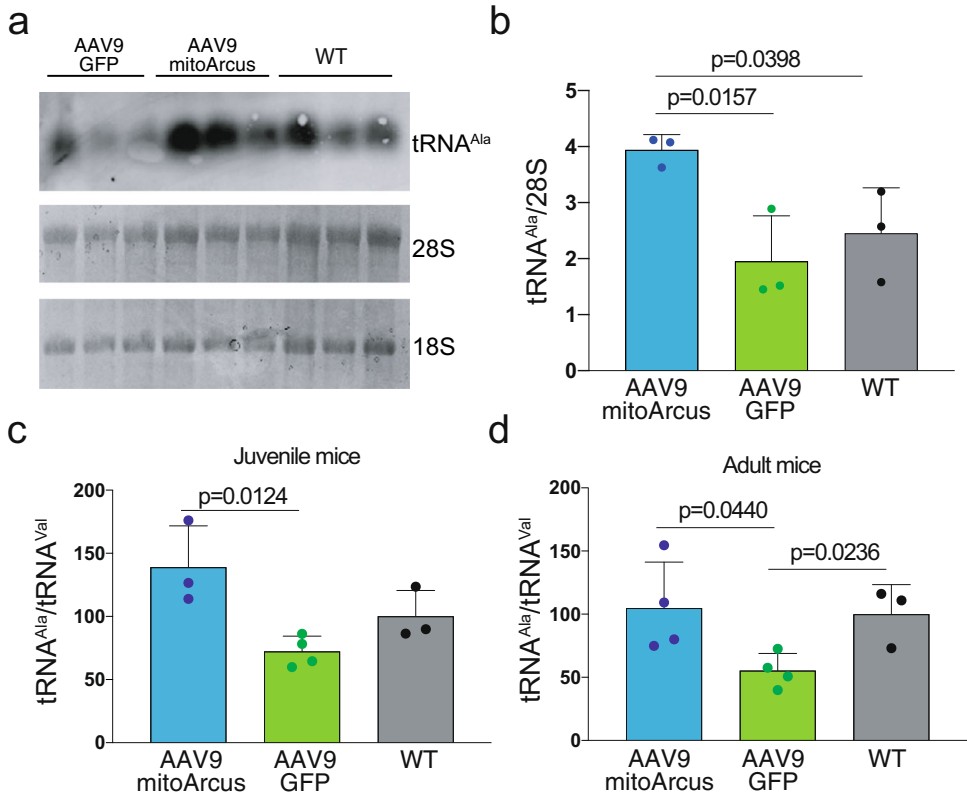

**Fig. 5 mitoARCUS-induced increase in mt-tRNA$^{Ala}$ in the liver. a** Northern blot analysis of juvenile mouse liver 24 weeks PI probed for mt-tRNA$^{Ala}$ and total RNA loading (28S and 18S). **b** Quantification of mt-tRNA$^{Ala}$ (from panel a) normalized to 28S rRNA ($n = 3$). Riboruler high-range molecular marker was used, but the lowest marker was 200 nt, whereas the tRNA are ~75 nt. **c** Quantification of mt-tRNA$^{Ala}$ by qPCR compared to levels of mt-tRNA$^{Val}$ in juvenile mouse liver 24 weeks PI. RNA samples from AAV9-mitoARCUS-treated animals compared to AAV9-GFP controls and WT liver samples ($n = 3$ with the exception of GFP $n = 4$). **d** Quantification of mt-tRNA$^{Ala}$ compared to levels of mt-tRNA$^{Val}$ in adult mouse liver. RNA samples from AAV9-mitoARCUS-treated animals compared to AAV9-GFP controls and WT liver samples, at 24 weeks PI ($n = 4$ with the exception of WT $n = 3$). Data are mean ± SEM. Statistical analysis was performed using two-tailed Student's $t$-test.

tRNA$^{Ala}$ mice[20,32]. We determined the mt-tRNA$^{Ala}$ levels in the liver of juvenile mice 24 weeks PI by northern blotting. Results showed an increased amount of mt-tRNA$^{Ala}$ in AAV9-mitoARCUS-injected animals compared to AAV9-GFP-injected controls when normalized to 28S rRNA (Fig. 5a, b). We further used quantitative PCR (qPCR) to determine the ratio of mt-tRNA$^{Ala}$ to mt-tRNA$^{Val}$. These results confirmed the northern blotting results (Fig. 5c, d). By qPCR, we found that AAV9-mitoARCUS-treated juvenile animals had significantly higher levels of mt-tRNA$^{Ala}$ compared to controls (AAV9-GFP, Fig. 5c). Adult animals injected with AAV9-mitoARCUS also had increased mt-tRNA$^{Ala}$ levels in the liver compared to controls (AAV9-GFP) (Fig. 5d).

## Discussion

Currently, most mitochondrial disorders have no treatment and care is only palliative. The ability to shift mtDNA heteroplasmy is a promising treatment approach for severe heteroplasmic mitochondrial disorders. This can be done by decreasing levels of mutant mtDNA to below disease threshold levels, to ameliorate phenotypes.

Many DNA-editing enzymes have been used to shift heteroplasmy, but they have their pitfalls in terms of potential clinical use. MitoZFN's and mitoTALENs have a heterodimeric architecture, making packaging into viral vectors difficult, requiring that each monomer is packaged into separate viral vectors. The CRISPR-Cas9 system is not appropriate for mtDNA

modification, because mitochondria do not have an RNA import mechanism[33]. More recently, a base editor DdCBE was shown to edit cytosines preceded by thymidines[34]. However, the sequence requirements limit its potential use at this time[35]. It has, however, recently been used to edit mtDNA in mouse embryos[36]. MitoARCUS overcomes disadvantages that mitoZFN and mito-TALENs present with size and viral packaging. As recombinant AAV has a packaging limit of up to 4.5 kb[37], mitoARCUS can be packaged into a single viral vector, which avoids problems with dilution of titer and can be injected at higher doses[25–27].

MitoARCUS-induced heteroplasmic shifts of up to 60% in cell culture experiments after 24 h and those changes were consistent for up to 3 weeks. This shift was enough to see an improvement of mitochondrial function in mitoARCUS-treated cells, such as the increase in OCR when compared to controls transfected with pLenti-GFP and untransfected cells. In cell-sorting experiments, a small shift in heteroplasmy was observed in "Black" cells, because we co-transfected cells with a mix of mitoARCUS and GFP-expressing plasmid (2 : 1 ratio). Although this increases the odds that "Green" cells co-express the mitoARCUS, some cells are expected to internalize mitoARCUS but not the GFP plasmid.

Cells treated with mitoARCUS showed a depletion of total mtDNA levels 24 h after transfection, but levels quickly recovered to normal over 3 weeks' time. As mitochondrial copy number is tightly regulated, transient depletion of mtDNA was recovered by the replication of residual mtDNA molecules to bring mtDNA levels back to normal[38]. Although we cannot rule out that some

WT mtDNA were cleaved by the nuclease, the fast recovery indicates that it is specific enough to promote a robust elimination of the mutant mtDNA.

When tested in vivo, AAV9-mitoARCUS effectively shifted heteroplasmy in several of the analyzed tissues of juvenile animals, with no depletion in total mtDNA levels at 6, 12, or 24 weeks. The more robust heteroplasmy shifts in juvenile animals compared to adult, treated animals are not surprising, as this was previously observed before in gene therapy studies[39] and is likely due to the young immune system being more receptive to the AAV therapy. However, differences in viral titer could also play a role, as juvenile animals were injected with 1.5× more vgs/kg of AAV9-mitoARCUS than adults. Mice at 2.5 weeks roughly correspond to a 2-year-old human and mice at 6 weeks old to a 20-year-old human[40]. These results suggest that this therapeutic approach could be more effective in younger patients. Also, fine-tuning dosing of AAV9-mitoARCUS could prevent a mtDNA depletion, while still inducing a heteroplasmic shift large enough to improve mitochondrial function, as was previously shown when mitoZFN were used at a lower dose[16]. In addition, the increase in mt-tRNA[Ala] levels in experimental animals further indicated that heteroplasmy shift in vivo can induce biochemical changes.

Mutant mtDNA in the liver was almost completely eliminated 6 weeks after injection of the AAV9-mitoARCUS. However, the expression of mitoARCUS in the liver at this time point was relatively low. As AAV9 is highly hepatotropic[28], AAV9-mitoARCUS likely transduced the liver strongly, but very early after injection, shifting heteroplasmy in the liver to nearly all wild-type mtDNA. As the liver is a highly metabolically active tissue, AAV9-mitoARCUS may have been mostly flushed out before the 6-week time point. On the contrary, other post-mitotic tissues such as the skeletal muscle, maintained expression through 24 weeks after injection of AAV9-mitoARCUS. This hypothesis was confirmed by analyzing animals 5 and 10 days after injection, which showed a strong mitoARCUS expression in the liver, but not in other tissues.

Although AAV9 has been shown to have tropism in a wide array of tissues[28,41,42], we did not find a strong expression in the central nervous system, which agrees with our previous experience with AAV9[17,18,20]. Curiously, we saw expression of AAV9-mitoARCUS in the heart, but with a small effect on heteroplasmy. We do not have an explanation for the cardiac resistance to mtDNA heteroplasmy change, but some cell types within the heart may be better transduced with AAV9 than others.

There has been an increased interest in recent years in the development of new AAV serotypes and their tissue specificity[43–47], and how it varies in different hosts. This is the case of a recent study where authors found that the genetic background of mice greatly affected the efficacy and specificity of transduction using the same AAV serotype[48,49]. The hope is that newly engineered strains of AAV and tissue-specific promoters will be able to overcome these barriers[50].

In conclusion, mitoARCUS is a promising tool for eliminating mutant mtDNA. Its compact size, monomeric structure, ability to recognize new target sequences, and ability to produce large shifts with minimal negative effects suggest that it could be further developed for in vivo applications.

## Methods

**MitoARCUS construct design**. Precision BioSciences used in silico predictions and directed evolution methods to re-engineer I-CreI, to recognize the m.5024C>T point mutation as described[26,27]. A peptide linker was used to fuse both homodimers into a monomeric structure[26,27]. The specificity of the mitoARCUS was

initially confirmed in CHO cells using an engineered split-GFP assay as described in Fig. 1a.

**Animal model and fibroblasts**. We used a heteroplasmic mouse model carrying a m.5024C>T point mutation in the tRNA[Ala] gene. This mouse model has been previously characterized[20,32,51] and present with a mild cardiomyopathy at 2 years of age, as well as reduced mt-tRNA[Ala] levels when mutation levels are >50% in the tissue. MEFs derived from this mouse were characterized for the levels of mutant mtDNA and immortalized with the E6–E7 gene of the human papilloma virus[20]. Wild-type animals and WT lung fibroblast cells used as controls were derived from mice with C57BL/6J background. To produce an MEF line with high levels of mutation, a mitoTALEN targeting WT mtDNA was designed. MEFs were transfected with the mitoTALEN, sorted, and clones were grown and tested for high levels of mutant mtDNA[20].

We have complied with all relevant ethical regulations for animal testing and research. The work with mice was approved by the University of Miami Institutional Animal Care and Use Committee.

**Cell culture, transfection, and sorting**. Heteroplasmic MEFs were transfected using GenJet DNA In Vitro Transfection Reagent (Ver. II) (SL100489, SignaGen Laboratories) using the manufacturer's protocols. We transfected cells plated in a T75 flask at 80% confluence with 30 μg plasmid total, in a 2:1 ratio of mitoARCUS CF or CSF plasmid (20 ug) to GFP plasmid (10ug). Twenty-four hours after transfection, sorting was performed using FACS Aria IIU, gating on single-cell fluorescence using a 488 nm laser and 505LP, 530/30 filter set for GFP expression. Cells were sorted into Black populations (no GFP expression) and Green populations (GFP expression). Untransfected control cells were also passed through the cell sorter in order to expose cells to the same conditions[18,20,21]. HeLa cells (CCL-2 Cells, ATCC) and HEK293T (CRL-3216, ATCC) were used.

**Immunofluorescence analysis**. HeLa cells were plated onto coverslips and transfected with mitoARCUS for 24 h. Cells were incubated for 1 h at 37 °C with 200 nM MitoTracker Red CMXRos (M7512, Invitrogen), protected from light. Then, cells were fixed with 2% paraformaldehyde for 15 min at room temperature (RT). Cells were permeabilized with 0.2% Triton X-100 in phosphate saline buffer (PBS) for 2 min at RT. A 3% bovine serum albumin (BSA)/PBS solution was used to block for 1 h at RT. The primary antibody against Flag (F3165, Sigma) at a 1:200 concentration in 2% BSA/PBS was incubated for 1 h at RT. After washing, cells were incubated with the secondary antibody: Alexa Fluor 488 goat anti-rabbit IgG (A-11008, Invitrogen) at a 1:200 concentration in 3% BSA/PBS for 1 h at RT, protected from light. Coverslips were then washed with PBS and mounted onto slides using a 4′,6-diamidino-2-phenylindole-containing mounting medium (Everbrite Mounting Medium, Biotium). Images were captured using a Zeiss LSM510 confocal microscope[11,20].

**AAV9 preparation and administration**. The mitoARCUS was cloned into an AAV2/9 plasmid and sent to the University of Iowa Viral Core Facility, which produced virus with $1.6 \times 10^{13}$ vg/ml titer for AAV-mitoARCUS and $5.9 \times 10^{13}$ vg/ml of AAV9-GFP. Both juvenile (2.5 weeks of age) and adult mice (6 weeks of age) were injected retro-orbitally as described[20,30]. Juvenile mice received $6.67 \times 10^{13}$ vgs/kg of AAV9-mitoARCUS or $6.15 \times 10^{13}$ vgs/kg of AAV9-GFP into the left retro-orbital sinus. Adults received $4.0 \times 10^{13}$ vgs/kg of AAV9-mitoARCUS or $3.69 \times 10^{13}$ vgs/kg AAV9-GFP into the left retro-orbital sinus. Control animals were injected with similar titers of AAV9-GFP, respective of age-matched experimental animals. Injections were carried out using a 31 G needle with syringe (08290-8438-01, BD). Toe biopsies were collected at 6 days of age, to determine base heteroplasmy levels. At 6, 12, and 24 weeks PI, mice were anesthetized with Ketamine and Xylazine, and were perfused with PBS. The heart, TA, quadriceps, gastrocnemius, kidney, liver, and spleen were collected. Samples were flash frozen in liquid nitrogen and then stored at −80 °C until further use. All animal procedures were approved by the University of Miami Animal Care and Use Committee.

**MitoARCUS expression**. Total protein homogenate was prepared from flash-frozen tissues and protein was quantified using DC protein assay (5000116, BioRad) according to the manufacturer's instructions. Forty-micrograms of total protein per sample was run in 10% Mini-PROTEAN TGX Stain-Free Protein Gels (4568034, BioRad), then transferred onto polyvinylidene difluoride membranes (1620260, BioRad) using the TransBlot Turbo system (1704155, BioRad) according to the manufacturer's instructions. TGX Stain-Free Protein Gels allow the visualization of total protein loading through gel activation with the BioRad Chemidoc system. Blots were blocked for 1 h at RT with 5% milk. Antibodies used were mouse monoclonal Flag (F3165, Sigma) (1:1000), mouse monoclonal GFP (75-131, UC Davis) (1:1000), mouse monoclonal MTCO1 (ab14705, Abcam) (1:1000), mouse monoclonal NDUFB8 (ab110242, Abcam) (1:750), mouse monoclonal Tubulin (T9026, Sigma) (1:20,000), rabbit polyclonal Caspase-3 (#9662, Cell Signaling) (1:1000), and mouse monoclonal PCNA (PC10 #2586, Cell Signaling) (1:2000). Secondary antibody was IgG horseradish peroxidase-linked mouse (7076, Cell Signaling) (1:5000) or rabbit (7074, Cell Signaling) (1:5000).

The primary antibody was incubated overnight at 4 °C and secondary antibody incubated for 1 h at RT. Membranes were developed with SuperSignal West Pico chemiluminescent substrate (34080, Thermo Scientific) and were imaged in the BioRad Chemidoc imager. BioRad Image Lab (version 6.0.1) was used to analyze blots[11,20].

**DNA extraction, quantification by "last-cycle HOT" PCR, and RFLP.** Total DNA was extracted from flash-frozen tissues using phenol–chloroform and from FACS-sorted cells using the NucleoSpin Tissue XS kit (740901.50, Takara). DNA concentration was determined spectrophotometrically (BioTek Synergy H1 hybrid). Levels of the m.5024C>T mutation were determined by "last-cycle hot" PCR[52], wherein the last cycle of the PCR is run using radioactively labeled nucleotides. This method removes interference from heteroduplexes formed by previous melting and annealing steps by only allowing visualization of nascent amplicons. PCR amplicons were obtained with the following primers: F-5′-CCACCCTAGCTATCATAAGCACA-3′ and B-5′-AAGCAATTGATTTG-CATTCAATAGATGTAGGATGAAGTCCTGCA-3′[20]. RFLP analysis was done by digesting amplicons with PstI-HF (R3140S, New England BioLabs), which digests the WT mtDNA but not the mutant mtDNA carrying the m.5024C>T point mutation. After digestion, products were run in a 12% polyacrylamide gel and signal was detected using the Cyclone phosphor-imaging system (Perkin Elmer) and OptiQuant software Version 5.0 (Perkin Elmer)[20].

**qPCR to determine total mtDNA levels.** To determine the total levels of mtDNA present in samples, we performed qPCR using TaqMan reagents (PrimeTime Std qPCR Assay, Integrated DNA Technologies)[20]. Samples were run on a BioRad CFX96/C1000 qPCR machine. Comparative cycle threshold (Ct) method was used to determine relative reads and total mtDNA levels were determined by comparing mtDNA (ND1 and ND5) to genomic DNA (18S). The following primer/probe sets were used[20]:

*mtDNA. ND1* = Forward: 5′-GCC TGA CCC ATA GCC ATA AT-3′ (NC_005089; mtDNA 3282-3301); reverse: 5′-CGG CTG CGT ATT CTA CGT TA-3′ (mtDNA 3402-3383). Probe: 5′-/56-FAM/TCT CAA CCC/ZEN/TAG CAG AAA CAA CCG G/3IABkFQ/-3′ (mtDNA 3310-3334).

*ND5* = Forward: 5′-CCC ATG ACT ACC ATC AGC AAT AG-3′ (mtDNA 12432-12454); reverse: 5′-TGG AAT CGG ACC AGT AGG AA-3′ (mtDNA 12533-12514). Probe: 5′-/5TET/AGT GCT/ZEN/GAA CTG GTG TAG GGC/3IABkFQ/-3′ (mtDNA 12482-12458).

*Genomic DNA. 18s* = Forward: 5′-GCC GCT AGA GGT GAA ATT CT-3′ (RefSeq NR_046233.2; chr17:39984253-39984272); reverse: 5′-TCG GAA CTA CGA CGG TAT CT-3′ (RefSeq NR_046233.2; chr17:39984432-39984412). Probe: 5′-/5Cy5/AAG ACG GAC CAG AGC GAA AGC AT/3IAbRQSp/-3′ (RefSeq NR_046233.2; chr17:39984285-39984305).

**Oxygen consumption rate.** OCR was measured using a Seahorse XFp Extra-cellular Flux Analyzer (Seahorse Bioscience). The day prior to the assay, cells were seeded at a density of 20,000 cells/well in wells B-G (wells A and H contained media only). The XFp sensor cartridge was calibrated with calibration buffer overnight at 37 °C. The following day, cell culture medium was replaced with low-buffered Seahorse medium supplemented with 10 mM glucose, 1 mM pyruvate, and 2 mM glutamine, and incubated for at least 1 h at 37 °C. Measurements of endogenous respiration were measured following each addition of 1 μM oligo-mycin, 0.5 μM Trifluoromethoxy carbonylcyanide phenylhydrazone (FCCP), and 1 μM rotenone plus antimycin A. Results were normalized to μg protein per well after the Seahorse run and protein was quantified using DC protein assay[21].

**RNA extraction, northern blot analysis, and quantification of mt-tRNA's.** RNA was isolated from flash-frozen tissues with TRIzol (Ambion) following the man-ufacturer's standard protocols. Samples were treated with DNase (AM1907, Invi-trogen) prior to spectrophotometric quantification. Northern blot analysis was done by running 4 μg total liver RNA per sample in a 1.2% agarose gel containing 20% formaldehyde and 1× 3-morpholinopropane-1-sulfonic acid (MOPS). Gel was run in 1× MOPS solution, first 80 V for 15 min, then 120 V for 2.5 h. At this point, the gel was stained with ethidium bromide (EtBr) to visualize total RNA loading. The gel was then washed two times for 10 min in water to remove EtBr. RNA was transferred overnight onto a nylon membrane (Amersham Hybond-NX, #RPN203T). Transcripts of interest were detected with non-radioactive biotiny-lated probes overnight at 50 °C. The following day, membrane was washed and signal was detected with IRDye 800CW Streptavidin (926-32230, Li-COR)[20]. We used a biotinylated probe to detect mitochondrial tRNA[Ala] by northern blotting: 5′-[Btn]GACTTCATCCTACATCTATTG-3′.

The levels of mt-tRNA[Ala] and mt-tRNA[Val] were detected using Custom TaqMan Small RNA Assay (4398987, ThermoFisher) as per the manufacturer's directions. Relative levels of mt-tRNA[Ala] were calculated by dividing Ct values by mt-tRNA[Val] Ct values.

**Detection of nuclear off-targets.** This assay is a modification of GUIDE-seq[53], known as "oligo capture," which is more sensitive in detecting ARCUS-induced DSBs[31]. FL83B mouse cells were electroporated with mitoARCUS and analyzed by oligo capture 2 days after transformation. Five C57BL6J nuclear genomic sequences were identified as putative off-target sites for this nuclease, as indicated in the table below. mitoARCUS-injected heteroplasmic mouse DNA samples (TA muscle and liver) were then analyzed for the presence of indels at these five nuclear genome sites. Site 1: Chromosome 2 (5′-GTAAGGATAGTAAGTCTTCCAA-3′); Site 2: Chromosome 2 (5′-TCAAGGATAGCACGTCTCCAAC-3′); Site 3: Chromosome 5 (5′-TTAAGGATGGTAAGACAGTATC-3′); Site 4: Chromosome 6 (5′-CTAAG GTTAGTAAGTATTCAAC-3′); Site 5: Chromosome 2 (5′-TTCAGGATGGCA CGTCTTCATC-3′). The primers used to amplify these potential off-target sites were as follows: Site1F 5′-cgcccccagcattgaagctg/Site1R tgcatggggccttagatttgc-3′; Site2F 5′-acagggagagactgaaccatcacag-3′/Site2R 5′-aggggaacaaaatatctagtgggtctag-3′; Site3F 5′-caaaactgaccgatcgatcctctg-3′/Site3R 5′-agatcacacacatgagccaacat-3′; Site5F 5′-cctcaagtgagcaggcacttacg-3′/Site5R 5′-cgagagttcactatgagtcagccatg-3′.

**Statistical analysis.** All data analysis was performed using GraphPad Prism 7 and 8. All statistics are presented as mean ± SEM. Pairwise comparisons were per-formed using the unpaired two-tailed Student's $t$-test. Comparisons between >2 groups were done by one-way analysis of variance. $P$-values of ≤0.05 were con-sidered significant. $N = 4$ mice were injected per each condition (treated vs. con-trol, and each time point). All measurements were taken from distinct samples.

**Reporting summary.** Further information on research design is available in the Nature Research Reporting Summary linked to this article.

## Data availability

All data and raw figures are available upon request. The only exception is the sequence of the mitoARCUS, which is under patent protection. However, it may be shared under appropriate MTAs, as detailed below: (A) Acceptance of a non-exclusive, non-transferable, non-sublicensable, and non-commercial research license during the term of the MTA that covers the use (but not the design or manufacture) of the ARCUS nucleases, solely to use them in the recipient's laboratories at the recipient's place of business to carry out a study as set forth in the MTA. (B) The licenses above will not include the right for Recipient to have any third party use any materials (including ARCUS nuclease sequence information) outside of the study and recipient shall not: (i) analyze, use (except as provided under this Agreement for the study), modify, or reverse engineer the materials (including ARCUS nuclease sequence information) in any way; or (ii) file any patent applications or seek other forms of statutory protection on the materials (including ARCUS nuclease sequence information), or any inventions conceived, discovered, developed, or otherwise made in the course of performing the study, or disclose any information or data gained from the materials (including ARCUS nuclease sequence information) or the study without the prior written consent of Precision. (C) ARCUS sequence information and all results derived from the materials or from the performance of the study shall be deemed to be confidential information of Precision. Recipient may use ARCUS sequence information and such results solely in furtherance of the study and for no other purposes. Any public disclosure or publication of ARCUS sequence information or results shall not be permitted without the prior written consent of Precision. Source data are provided with this paper.

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

## Acknowledgements

This work was supported primarily by the Muscular Dystrophy Association and the National Institute of Health Grants 5R01EY010804. Additional support was received from NIH grants 1R01NS079965, 1R01AG036871, and the CHAMP Foundation.

## Author contributions

U.Z. performed most of the experiments and analyzed the results. S.R.B. performed mouse injections and helped interpret the results. J.S. and D.J. designed the ARCUS meganuclease that G.T. built. W.S. performed the initial ARCUS specificity analysis. C.V.P. developed a cell line with high levels of mutant mtDNA. J.S. characterized and provided the mouse model. D.J. supervised the project at Precision. C.T.M. supervised the overall project and wrote the first draft with U.Z. All authors participated in editing of the manuscripts.

## Competing interests

The authors declare the following competing interests: W.S., J.S., G.T., and D.J. work at Precision BioSciences, which is researching and developing ARCUS technology, and may research, develop, and commercialize mitochondria-targeted ARCUS nucleases in the future. C.T.M., D.J., G.T., and W.S. are named in a joint patent application (Patent applicant- Precision BioSciences, Inc. and the University of Miami; application number: 63/178,269; status: pending) for mitochondria-targeted ARCUS nucleases with Precision BioSciences. Specific aspect of manuscript covered in patent application: ARCUS nuclease sequences and compositions for targeting mitochondria, including delivery vectors; Methods of modifying cells, Treating mitochondrial conditions in vivo using the disclosed ARCUS nucleases. All other authors declare no competing interests.
