## [Peer Review File · Nature Communications]

Reviewers' Comments:

Reviewer #1:

Remarks to the Author:

Brief Summary:

In this manuscript, the authors showed that mitoARCUS can be targeted to mitochondria where the monomeric protein induces shift of mtDNA heteroplasmy toward WT mtDNA-enriched normal populations. To show feasibility of their approach, they used an in vitro system in which cultured cells with a mitochondrial genetic mutation were treated with mitoARCUS-encoding plasmid. They successfully showed enrichment of normal mitochondrial population especially in GFP-positive cells. Finally, they provided evidence that mitoARCUS is effective to shift heteroplasmy in an m.5024C>T model mice when injected intravenously using AAV9 as a vector. In particular, the liver and skeletal muscle were prominently restored as assessed by the measurement of mt-tRNAAla levels. From these results, they concluded that mitoARCUS provide a treatment option for mitochondrial genetic disorders that have had no available treatments hitherto.

Overall Impression:

The authors employed a straightforward strategy in this manuscript to treat mtDNA heteroplasmy, and it is acknowledgeable that mitoARCUS can provide a feasible treatment option that has multiple advantages compared to ZFN or TALEN technology, in terms of delivery and efficacy. Though not totally novel, this study is somehow interesting in that mitoARCUS showed efficacy in several tissues using an in vivo mouse model. However, the results are quite limited to support that mitoARCUS is a powerful therapeutic option for mitochondrial genetic disorders. In particular, the authors paid no attention to specificity and safety issues on mitoARCUS treatment together with several analytical validity issues. Regarding this, several major issues are presented as follows together with minor points.

Major points:

1. Every restriction enzyme and genome editor innately show some degree of non-specific cleavages. And, the specificity varies under different physical, chemical and biological conditions. However, no one can assess from the presented data how specific mitoARCUS is. The authors are suggested to gauge the specificity in vitro system where both WT and Mut mtDNA are cleaved by recombinant mitoARCUS and to show what's the degree of cleavage of WT DNA and what's the ratio of cleavage of Mut DNA over WT one.
2. Mitochondrial DNA occupies only a small fraction of total DNA in cells because most of DNA comes from genomic DNA. Thus, the authors need to show shift of heteroplasmy in isolated mitochondrial samples at least for cell experiments.
3. They showed efficacy at the tissue level. However, no data are available at an individual organism level. Is there any difference in body weight between mitoARCUS and GFP transfected individuals? How about muscular strength? Is there any change in liver function? Your provision of such data would be required considering the high quality journal of Nat. Commun.
4. For the liver tissues, the expression of mitoARCUS was not observed on western blot analyses. The authors attributed the results to a prompt turnover in the liver cells. However, there is a possibility that the mtDNA-cleaved liver cells may undergo apoptotic pathway. To rule out this possibility, they need to show data supporting no occurrence of damage in liver cells, or at least, western blot analysis right after AAV injection, that is, within 1-2 days after injection.
5. It is estimated that the authors presented data on total mtDNA that showed no significant alterations after mitoARCUS administration. For this, they provided quantitative analysis data through qRT-PCR using genomic DNA as a control. In relation to toxicity issues mentioned in point #4, if cells undergo apoptotic pathway, there would be no change in the ratio of mtDNA/genomic DNA because the corrected cells just disappear. This may possibly occur because most cellular apoptosis is linked to the mitochondrial stress. The cleavage of mtDNA can be a trigger for cellular apoptosis.
6. It is likely that the quantification of MUT-to-WT shift is achieved by band intensities on gel images. For more quantitative analysis, I suggest the quantification using ddPCR at least for the

most important Figure.

7. In Fig. 2g, the authors investigated the oxygen consumption rate for both mitoARCUS and GFP-treated cells. The unit was expressed as pmolO₂/min/ug protein. It is probable that MUT mitochondria have defects both on oxygen consumption and protein production. Then, the net value would not be changed. Conversely, the shifted cells would show increased O₂ consumptions, but also increased protein levels at the same time. Would it possible to test the OCR as the unit of pmolO₂/min/number of mitochondria?

Minor points:

1. In Fig. 2, Fig.2g and Fig.2f were mistakenly numbered.
2. The method employed in Fig.1c is immunofluorescence, not immunocytochemistry.
3. In Fig. 3b and Fig.4b, the results were duplicated in multiple mice or one mouse? If it is derived from one mouse, it is not enough to perform statistical analysis. If multiple mice, the number of individuals should be presented.

Reviewer #2:

Remarks to the Author:

This manuscript describes an innovative approach that shows promise for potential treatment of mtDNA disease. It uses a mitochondrial-targeted meganuclease (mitoARCUS) to attempt to eliminate mutant mtDNA in cultured MEFs from a heteroplasmic mouse model as well as by direct intravenous delivery via an AAV9 vector to heteroplasmic mice. The authors note that existing tools that seek to cleave and eliminate mutant mtDNA (mtRE, mtZFN, mitoTALENs and mitoTev-TALE) or perform gene editing of mtDNA (cytidine deaminase) have limitations related to large size, heterodimeric structure, limited targeting ability or effectiveness that may limit their clinical utility. They have collaborated with a biotech company to develop and provide strong proof of principle that their mitoARCUS approach shows promise of in vivo efficacy. The work is thus of interest to the fields of mitochondrial disease, gene therapy and more broadly. It will influence thinking in the mitochondrial disease field about potential approaches to gene therapy and could eventuate in a treatment for some patients who currently lack any effective therapies.

While of high interest the work does have a number of shortcomings or ambiguities that could be addressed to improve its significance, as follows.

1) mitoARCUS and vector generation.

The description of the generation of the candidate meganuclease was described very briefly with the Methods referring only to references 26, 27, which are both patent applications. I appreciate there may be IP concerns here but if not described previously in peer-reviewed publications then I suggest a more detailed description is needed, preferably including the mutation targeting sequence. It also needs better referencing for the mitochondrial localization sequences Cox8 and Cox8/Su9, which I believe have been described previously but are not cited.

2) Cell transduction efficiency.

Fig.1 provides no indication of what proportion of HeLa cells were transfected and it would be useful to clarify this for HeLa cells and the subsequent MEF studies where they co-transfect with a GFP plasmid and only report on the GFP-positive cells. The Nat Comms reporting summary sheet mentions green cells comprising up to 11-20% of the cell population but it is not clear if this relates to HeLa cells or MEFs or both. Most of the relevant cell studies are done on green cells that have taken up the GFP vector as well as the mitoARCUS vector, so one expects to see a substantial shift to wildtype mtDNA if successful, which appears to be the case. However, some results on negative/black cells that presumably comprise mostly cells that lack both vectors are surprising unless a substantial proportion are expressing sufficient nuclease to be able to reduce the heteroplasmy level of the bulk population. The apparent improvement in OCR data in black cells shown in Fig. 2g is surprising given one would expect that the transfection efficiency was low in these cells i.e., we expect a small proportion of cells are transfected and individual cells may show large changes in heteroplasmy shift but that would not be expected to be seen as marked

changes in heteroplasmy or improved OCR in the bulk cell population. The data seem more consistent with the idea that many cells have had a modest decrease in mutant load rather than a small proportion have had a marked improvement. This warrants clarification and single cell RNAseq may be the most appropriate way to do this. Please note that the labels for Fig.2f and Fig.2g should be swapped in the figure. They are currently not consistent with the figure legend or text.

3) Mouse experiments.

It is impressive that both juvenile (Fig.3) and adult (Fig.4) heteroplasmic mice injected with AAV9-mitoARCUS vectors show almost complete elimination of mutant mtDNA in liver within 6 weeks. This encouraging result is perhaps the most exciting aspect of the manuscript and perhaps not surprising since AAV9 has high tropism for liver. However, the juvenile data are surprising in that there is almost no detectable expression of the FLAG tag from the vector at 6 weeks. In the Discussion they suggest this could be due to AAV9-mitoARCUS transducing the liver strongly but being "mostly flushed out" before 6 weeks, by when the mutant mtDNA has been almost completely eliminated. This is plausible but given the potential significance of this highly efficient elimination of mutant mtDNA. I suggest they confirm their speculation by sacrificing some juvenile mice at earlier time intervals e.g., 1 week PI to compare expression levels and heteroplasmy shift. The data in Fig.5 offer strong reassurance that the heteroplasmic shift in liver is reflected by increased levels of mt-tRNA-Ala, although it would be helpful to clarify in the figure legend if the data in all panels are from 24 weeks PI. One other confusing aspect of the mouse data is that the authors describe collecting mouse toe biopsies at 6 days of age to determine base heteroplasmy levels but don't seem to have used that data anywhere. Instead they reference heteroplasmy shifts to data from brain with the rationale that brain showed negligible transfection with their AAV9 vector. Why did they not use toe biopsies as the reference for this?

4) Comments on potential limitations.

Some reviewers could downplay the significance of their data showing successful shifting of heteroplasmy by noting that most mtDNA point mutations do not impact heavily on liver function or noting that the improvements seen in three skeletal muscle types were not as marked when adult mice were treated. I would not support that view as the striking findings in juvenile tissue suggest to me that with the ever improving targeting efficiency of AAV vectors and demonstrated success of recent human AAV trials their approach offers strong prospects of being able to be targeted to multiple tissues in the next few years. Two other concerns that have been raised with strategies attempting to eliminate mutant mtDNA is that they will not work for homoplasmic mtDNA mutations and run the risk of causing mtDNA depletion in target tissues of heteroplasmic patients. While those are potential limitations, the authors rightly point out that mtDNA gene editing approaches are currently inefficient and unable to target some of the most common mtDNA mutations. They also report only modest if any reductions in total mtDNA levels in tissues from treated mice. Hence, the results described show impressive potential for mitoARCUS as a future treatment approach. In future studies it would be desirable to look for any evidence of focal pathology in addition to bulk tissue effects, although I would not require that for the current study. For example, hepatic mtDNA depletion frequently affects only a small proportion of cells initially, which can be detected by approaches such as electron microscopy or immunohistochemistry for OXPHOS subunits to identify subsets of hepatocytes with disrupted morphology or enzyme defects. Likewise, in future studies it would seem desirable to perform immunohistochemistry or another method to analyse distribution of GFP within tissues like heart to explain findings like the lack of heteroplasmy shift in heart despite apparently strong AAV9-mitoARCUS expression.

We thank the Reviewers for the comprehensive critique of our submission. We addressed their concerns below.

Reviewer 1:

Major points:

1. *“Every restriction enzyme and genome editor innately show some degree of non-specific cleavages. And, the specificity varies under different physical, chemical and biological conditions. However, no one can assess from the presented data how specific mitoARCUS is. The authors are suggested to gauge the specificity in vitro system where both WT and Mut mtDNA are cleaved by recombinant mitoARCUS and to show what’s the degree of cleavage of WT DNA and what’s the ratio of cleavage of Mut DNA over WT one.”*

We believe that our GFFP assay in CHO cells is very sensitive and showed the relative specificity in cleavage between WT and mutant target sites within a cellular context (Figure 1a). We agree with the Reviewer that at a certain concentration, the nuclease is likely to cleave the WT sequence as well. However, the GFFP assay used in CHO cells shows the relative specificity in a cellular context where the levels of expression are limited by the vector and mode of transduction. The *in vitro* assay suggested by the Reviewer would support assays using very high levels of the nuclease, but we would not be able to directly extrapolate those to *in vivo* assays.

In addition, we have now included a new analysis for off-target cleavage in the nucleus. Even though we do not observe the enzyme in the nucleus, we thought it would be a valuable addition to the study. We examined 5 potential off-target loci in the nucleus and found no evidence of cleavage (described in results).

2. *“Mitochondrial DNA occupies only a small fraction of total DNA in cells because most of DNA comes from genomic DNA. Thus, the authors need to show shift of heteroplasmy in isolated mitochondrial samples at least for cell experiments.”*

We are not exactly sure what the Reviewer means. Our assay for heteroplasmy is specific for mtDNA.

If the Reviewer meant that we should promote the shift in heteroplasmy in isolated mitochondria, this is not possible as there is no accepted methodology to transform mitochondria directly. If the Reviewer was concerned with PCR amplification from non-mitochondrial sources, we have controlled that by attempting to amplify the target using mouse cells without mtDNA and found no amplifications. We also have cell lines with different levels of mutant mtDNA that are accurately quantified using our “last cycle hot” PCR approach (see below).

3. *“They showed efficacy at the tissue level. However, no data are available at an individual organism level. Is there any difference in body weight between mitoARCUS and GFP transfected individuals? How about muscular strength? Is there any change in liver function?”*

We did not quantify differences in muscular strength in treated and control mice. However, mice were closely monitored and did not show any overt phenotypes, moving around the cage as

AAV-GFP injected mice. We did collect weight data, which showed no differences, and have now included it in Supplementary Figure S1. Regarding liver function, following the Reviewers suggestions, we performed new experiments to look at shorter time points. At 5 and 10 days after injections, liver was the only tissue showing changes in mtDNA heteroplasmy. We did not observe evidence of liver damage by analyzing apoptosis markers (Supplementary Figures S2 and S3). Moreover, no abnormalities were observed in H&E slides (Supplementary Figure S4).

4. *“For the liver tissues, the expression of mitoARCUS was not observed on western blot analyses. The authors attributed the results to a prompt turnover in the liver cells. However, there is a possibility that the mtDNA-cleaved liver cells may undergo apoptotic pathway. To rule out this possibility, they need to show data supporting no occurrence of damage in liver cells, or at least, western blot analysis right after AAV injection, that is, within 1-2 days after injection”.*

Following the Reviewer’s advice, we repeated the injections and analyzed liver at 5 and 10 days PI. We observed mitoARCUS expression in liver, as expected (Supplementary Figure S2 and S3). AAV9 transduction is known to not cause liver damage or hepatocyte apoptosis {Chen, 2015 #1}. Accordingly, the new experiments showed no evidence of apoptosis or liver regeneration. We also did not detect a depletion of mtDNA, which would occur if the mitoARCUS was not specific, even at these early time points.

5. *“It is estimated that the authors presented data on total mtDNA that showed no significant alterations after mitoARCUS administration. For this, they provided quantitative analysis data through qRT-PCR using genomic DNA as a control. In relation to toxicity issues mentioned in point #4, if cells undergo apoptotic pathway, there would be no change in the ratio of mtDNA/genomic DNA because the corrected cells just disappear. This may possibly occur because most cellular apoptosis is linked to the mitochondrial stress. The cleavage of mtDNA can be a trigger for cellular apoptosis. “*

The distribution of mutant and WT mtDNA among tissues and hepatocytes is relatively homogeneous. We have now analyzed markers of apoptosis and found no difference between mitoARCUS and GFP- injected livers.

6. *“It is likely that the quantification of MUT-to-WT shift is achieved by band intensities on gel images. For more quantitative analysis, I suggest the quantification using ddPCR at least for the most important Figure.”*

Heteroplasmy determination is performed using a technique known as “last cycle hot PCR”. Band intensities were measured in a phosphoimager, within the linear range. Radioactive nucleotides were incorporated into new strands only in the last cycle of PCR, avoiding the detection of heteroduplexes, which would behave as “uncut” in the RFLP assay. This method is quantitative and has been extensively used in the field since described in 1992 {Moraes, 1992 #2}.

7. *“In Fig. 2g, the authors investigated the oxygen consumption rate for both mitoARCUS and GFP-treated cells. The unit was expressed as pmolO₂/min/ug protein. It is probable that MUT mitochondria have defects both on oxygen consumption and protein production. Then, the net value would not be changed. Conversely, the shifted cells would show increased O₂*

consumptions, but also increased protein levels at the same time. Would it possible to test the OCR as the unit of pmolO₂/min/number of mitochondria?"

Our group has previously looked at defects of mitochondrial protein synthesis in heteroplasmic cells in culture carrying high levels of the mtDNA C5024T mutation and found no major decrease in mitochondrial protein production when compared to WT cells. Moreover, the mitochondrial protein synthesis is responsible for less than 5% of the total mitochondrial proteins, which are mainly encoded by nuclear DNA and synthesized in cytoplasmic ribosomes. As such, we believe that using our current measurement of pmolO₂/min/μg protein is reliable. Additionally, we plated the same number of cells/well for every experiment and the results were similar if normalized by cell number.

Minor points:

1. Correct label of Fig.2g and Fig.2f.

Done.

2. The method employed in Fig.1c is immunofluorescence, not immunocytochemistry.

Corrected as suggested.

3. In Fig. 3b and Fig.4b state the number of mice used.

Done (n=3-4).

Reviewer 2:

1. "The description of the generation of the candidate meganuclease was described very briefly with the Methods referring only to references 26, 27, which are both patent applications. I appreciate there may be IP concerns here but if not described previously in peer-reviewed publications then I suggest a more detailed description is needed, preferably including the mutation targeting sequence. It also needs better referencing for the mitochondrial localization sequences Cox8 and Cox8/Su9, which I believe have been described previously but are not cited. "

We have improved the description of the reagent and added references for the COX8 and COX8/Su9 mitochondrial localization sequences. We have also included the target sequence in the mtDNA, as requested by the Reviewer (first paragraph of Results).

2. "Fig.1 provides no indication of what proportion of HeLa cells were transfected and it would be useful to clarify this for HeLa cells and the subsequent MEF studies where they co-transfect with a GFP plasmid and only report on the GFP-positive cells. The Nat Comms reporting summary sheet mentions green cells comprising up to 11-20% of the cell population but it is not clear if this relates to HeLa cells or MEFs or both. Most of the relevant cell studies are done on green cells that have taken up the GFP vector as well as the mitoARCUS vector, so one expects to see a substantial shift to wildtype mtDNA if successful, which appears to be the case. However, some results on negative/black cells that presumably comprise mostly cells that lack both vectors are surprising unless a substantial proportion are expressing sufficient nuclease to

be able to reduce the heteroplasmy level of the bulk population. The apparent improvement in OCR data in black cells shown in Fig. 2g is surprising given one would expect that the transfection efficiency was low in these cells i.e., we expect a small proportion of cells are transfected and individual cells may show large changes in heteroplasmy shift but that would not be expected to be seen as marked changes in heteroplasmy or improved OCR in the bulk cell population. The data seem more consistent with the idea that many cells have had a modest decrease in mutant load rather than a small proportion have had a marked improvement. This warrants clarification and single cell RNAseq may be the most appropriate way to do this. Please note that the labels for Fig.2f and Fig.2g should be swapped in the figure. They are currently not consistent with the figure legend or text.”

These concepts have been clarified in the text and the reporting summary.

HeLa cells transfected in Figure 1 were only used for immunofluorescence, and therefore transfection efficiency is not critical. All cell sorting experiments (Figure 2) that were used in analysis of mtDNA depletion, heteroplasmy change, and oxygen consumption rates were performed in MEFs derived from the heteroplasmic mouse model. In these experiments, transfection efficiency of GFP-positive cells varied 11-20%. In these experiments, we transfected with twice the amount of mitoARCUS expressing plasmid than GFP expressing plasmid, increasing the odds that green cells would co-express the mitoARCUS. However, this ratio also allows for some cells internalizing the mitoARCUS and not the GFP plasmid. Also, we believe that the single cells PCR experiments are not necessary, as heteroplasmy change depends on levels of mitoARCUS expression, which will be different between cells, depending on the plasmid incorporation in each cell. In any case, reducing mutant load, even by a small percentage, would improve overall mitochondrial function.

We have corrected the Fig 2 legend.

3) “It is impressive that both juvenile (Fig.3) and adult (Fig.4) heteroplasmic mice injected with AAV9-mitoARCUS vectors show almost complete elimination of mutant mtDNA in liver within 6 weeks. This encouraging result is perhaps the most exciting aspect of the manuscript and perhaps not surprising since AAV9 has high tropism for liver. However, the juvenile data are surprising in that there is almost no detectable expression of the FLAG tag from the vector at 6 weeks. In the Discussion they suggest this could be due to AAV9-mitoARCUS transducing the liver strongly but being “mostly flushed out” before 6 weeks, by when the mutant mtDNA has been almost completely eliminated. This is plausible but given the potential significance of this highly efficient elimination of mutant mtDNA. I suggest they confirm their speculation by sacrificing some juvenile mice at earlier time intervals e.g., 1 week PI to compare expression levels and heteroplasmy shift. The data in Fig.5 offer strong reassurance that the heteroplasmic shift in liver is reflected by increased levels of mt-tRNA-Ala, although it would be helpful to clarify in the figure legend if the data in all panels are from 24 weeks PI. One other confusing aspect of the mouse data is that the authors describe collecting mouse toe biopsies at 6 days of age to determine base heteroplasmy levels but don’t seem to have used that data anywhere. Instead they reference heteroplasmy shifts to data from brain with the rationale that brain showed negligible transfection with their AAV9 vector. Why did they not use toe biopsies as the reference for this?”

Following the reviewer's suggestion, we produced more recombinant AAV9-mitoARCUS and AAV9-GFP and performed additional experiments. Young mice were injected and analyzed 5 and 10 days post injection. The results clearly show expression of mitoARCUS in the liver and already a significant elimination of mutant mtDNA (Supplementary Figure S2 and S3).

We have clarified in the Figure 5 legend that the data in all panels are from 24 weeks PI.

Toe biopsies were collected at 6 days to make sure the animal was heteroplasmic before injections. However, we used tissues that do not express mitoARCUS as controls because the quality of DNA was better. Furthermore, Supplementary Figure S2b and S3b demonstrate that heteroplasmy of Tails Before and After injection are the same, and levels of mutant mtDNA are similar to Brain.

4) *“Some reviewers could downplay the significance of their data showing successful shifting of heteroplasmy by noting that most mtDNA point mutations do not impact heavily on liver function or noting that the improvements seen in three skeletal muscle types were not as marked when adult mice were treated. I would not support that view as the striking findings in juvenile tissue suggest to me that with the ever improving targeting efficiency of AAV vectors and demonstrated success of recent human AAV trials their approach offers strong prospects of being able to be targeted to multiple tissues in the next few years. Two other concerns that have been raised with strategies attempting to eliminate mutant mtDNA is that they will not work for homoplasmic mtDNA mutations and run the risk of causing mtDNA depletion in target tissues of heteroplasmic patients. While those are potential limitations, the authors rightly point out that mtDNA gene editing approaches are currently inefficient and unable to target some of the most common mtDNA mutations. They also reported only modest if any reductions in total mtDNA levels in tissues from treated mice. Hence, the results described show impressive potential for mitoARCUS as a future treatment approach. In future studies it would be desirable to look for any evidence of focal pathology in addition to bulk tissue effects, although **I would not require that for the current study.** For example, hepatic mtDNA depletion frequently affects only a small proportion of cells initially, which can be detected by approaches such as electron microscopy or immunohistochemistry for OXPHOS subunits to identify subsets of hepatocytes with disrupted morphology or enzyme defects. Likewise, in future studies it would seem desirable to perform immunohistochemistry or another method to **analyze distribution of GFP within tissues like heart to explain findings like the lack of heteroplasmy shift in heart despite apparently strong AAV9-mitoARCUS expression.**”*

We agree with the Reviewer that delivery of therapeutic genes to desired tissues remains a challenge. It is also true that this therapy would not work for homoplasmic mitochondrial mutations.

Apoptosis markers were not increased in mitoARCUS expressing liver, even at earlier time points. Moreover, H&E histology staining on Liver samples done 5 and 10 days PI, showed no differences between injected animals, and age-matched non-injected animal (Supplementary Figure S4).

Finally, we agree with the Reviewer that the lack of heteroplasmy shift in heart certainly requires future exploration.

MILLER
SCHOOL OF MEDICINE
UNIVERSITY OF MIAMI

Department of Neurology
1095 NW 14th Terrace
Miami, Florida 33136
305-243-5858

Reviewers' Comments:

Reviewer #1:

Remarks to the Author:

Defects in mitochondrial tRNA genes are reported to cause several pathological phenotypes including loss of muscle, neurological defects, metabolic defects, etc.

In this manuscript, the authors injected AAV9-mitoArcus into mice with m.5024C>T tRNAAla. Depending on the relative ratio of MUT over WT, the mice may show various pathological phenotypes, which are expected to ameliorate by AAV treatment. However, the authors did not provide any sign of health improvement in vivo. Other than molecular signature, what's the improvement in health by AAV-mitoARCUS treatment? This may be related to the reason the authors used the serotype of AAV9 among others.

Reviewer #2:

Remarks to the Author:

The authors have performed additional experiments and clarified details that adequately address my concerns. Specifically:

1. While details of generation of mito meganucleases are still only described in patent applications, other details provided clarify the mutation targeting sequence and mitochondrial localization sequences.
2. My concerns about heteroplasmy correction in green vs black cells were addressed adequately by clarification of transfection efficiencies and the ratio of mitoARCUS to GFP plasmids.
3. Supp Figs S2 to S4 showing data on mice sacrificed at 5 and 10 days PI address my concerns about demonstrating early expression of mitoARCUS in liver and its subsequent loss, presumably related to ongoing cell division in juvenile liver.
4. My final major comment was really a comment about potential future avenues of research and did not require changes by the authors.

I had no particular concerns about responses to the other reviewer. My only comment would be that while quantitation of mtDNA heteroplasmy by last-hot cycle PCR is a bit "old school" now that high depth NextGen sequencing methods are widely used for this purpose, it is a robust, well validated technique that is fit for purpose for the studies in this manuscript.

The only minor issue I noted in the revisions was that in Supp Figs S2B and S3B, it would be helpful to label the RFLP bands as MUT and WT.

Response to Reviewers

Reviewer #1 (Remarks to the Author):

Defects in mitochondrial tRNA genes are reported to cause several pathological phenotypes including loss of muscle, neurological defects, metabolic defects, etc.

In this manuscript, the authors injected AAV9-mitoArcus into mice with m.5024C>T tRNA^{Ala}. Depending on the relative ratio of MUT over WT, the mice may show various pathological phenotypes, which are expected to ameliorate by AAV treatment. However, the authors did not provide any sign of health improvement in vivo. Other than molecular signature, what's the improvement in health by AAV-mitoARCUS treatment? This may be related to the reason the authors used the serotype of AAV9 among others.

Unfortunately, there are no mouse models with heteroplasmic mtDNA mutations that show a robust phenotype. In fact, there are very few mouse models with mtDNA mutations. The one used in this study is no exception. The mouse with a heteroplasmic C5024T mutation in the tRNA^{ALA} gene of mtDNA shows a mild phenotype only when the levels of mutation are very high, which is difficult to obtain by breeding. Even in these cases, very old mice show a very mild cardiomyopathy. Therefore, we showed a nuclease-dependent improvement in the molecular pathology, which is an improvement in tRNA alanine levels. We did show improvement in mitochondrial function in a cell model, which we could manipulate to be more than 95% mutant mtDNA.

Reviewer #2 (Remarks to the Author):

The authors have performed additional experiments and clarified details that adequately address my concerns. Specifically:

1. While details of generation of mito meganucleases are still only described in patent applications, other details provided clarify the mutation targeting sequence and mitochondrial localization sequences.
2. My concerns about heteroplasmy correction in green vs black cells were addressed adequately by clarification of transfection efficiencies and the ratio of mitoARCUS to GFP plasmids.
3. Supp Figs S2 to S4 showing data on mice sacrificed at 5 and 10 days PI address my concerns about demonstrating early expression of mitoARCUS in liver and its subsequent loss, presumably related to ongoing cell division in juvenile liver.
4. My final major comment was really a comment about potential future avenues of research and did not require changes by the authors.

I had no particular concerns about responses to the other reviewer. My only comment would be that while quantitation of mtDNA heteroplasmy by last-hot cycle PCR is a bit "old school" now that high depth NextGen sequencing methods are widely used for this purpose, it is a robust, well validated technique that is fit for purpose for the studies in this manuscript.

The only minor issue I noted in the revisions was that in Supp Figs S2B and S3B, it would be helpful to label the RFLP bands as MUT and WT.

We thank the Reviewer for the comments. We have labeled the Supplemental figures as requested.